# Evaluating InferVision's Computer-Aided Detection (CAD) algorithm for Tuberculosis (TB) screening, Lusaka, Zambia

**Paul Somwe** [id]*, **Minyoi Maimbolwa, Kanema Chiyenu, Mwansa Lumpa** [id],
**Mary Kagujje, Monde Muyoyeta**

Centre for Infectious Disease Research in Zambia (CIDRZ), Lusaka, Zambia

* paul.somwe@cidrz.org

## Abstract

The objective of this study was to evaluate the diagnostic performance of InferRead DR Chest for tuberculosis (TB) screening in a high HIV and TB burden setting. The study assessed the performance of InferRead DR Chest using anonymized chest X-ray images from an active TB case finding study in Lusaka, Zambia, for individuals aged 15 and older. The Xpert MTB/RIF or MTB culture was the composite reference standard. Performance was evaluated using the Area Under the Receiver Operating Characteristic Curve (AUC), and a binary classification point was selected where the sensitivity aligned with the WHO target product profile for TB screening tools. Of the 1,890 chest X-ray images that met the inclusion criteria, 91.5% of participants reported at least one TB symptom. The median age was 38 years (IQR: 29–47), and 1,186 (62.8%) were male. From the study sample, 449 participants (23.8%) reported a history of previous TB, and 704 (37.2%) were HIV positive. Among the analyzed images, 289 (15.3%) were classified as TB positive based on the composite reference standard test results. The overall area under the curve (AUC) was 0.81 (95% CI: 0.78–0.83). Among individuals with a history of previous TB and those who were HIV positive, the AUCs were 0.71 (95% CI: 0.63–0.79) and 0.77 (95% CI: 0.72–0.82), respectively. At a sensitivity of 90.3% (95% CI: 86.3%–93.5%), InferRead DR Chest achieved a specificity of 39.2% (95% CI: 36.8%–41.7%) at TB score cut point of 0.12. InferRead DR Chest had acceptable performance in our population. Additional training and piloting of InferRead DR Chest in this population is recommended.

## Introduction

Tuberculosis (TB) remains a global health challenge, with millions of new cases reported each year, resulting in substantial morbidity and mortality [1,2]. Accurate and timely diagnosis of TB is paramount not only for effective patient management but also for the prevention of disease transmission [3–5]. In this context, Computer-Aided

**Data availability statement:** All relevant data are within the paper and its Supporting Information files.

**Funding:** This study received support from the TB REACH initiative of the Stop TB Partnership, funded by the Government of Canada, grant number STBP/TBREACH/GSA/W5-26 (to MM). The funders had no role in study design, data collection and analysis, decision to publish, or preparation of the manuscript.

**Competing interests:** The authors have declared that no competing interests exist.

Detection (CAD) has emerged as a promising tool that can revolutionize TB diagnosis by offering automated and objective assistance to healthcare providers [6]. While traditional diagnostic methods have been the cornerstone of TB diagnosis, new technological advancements are paving the way for more efficient and accurate detection strategies [7,8]. The World Health Organization (WHO) recommends CAD for screening and triage purposes, but it cannot replace microbiological diagnostic tests such as nucleic acid amplification tests (NAAT) or culture, which are still required for a definitive TB diagnosis [9–11].

In 2021, the WHO recognized the potential of CAD technology [11]. This recognition signified a crucial step in the evolution of TB screening and highlights the growing importance of CAD software in improving the accuracy and efficiency of TB diagnosis [12]. This technology can be especially valuable when access to skilled radiologists is limited [13]. Furthermore, CAD technology can be deployed in various healthcare settings, including rural and underserved areas, extending the reach of TB diagnostic services. This is particularly relevant in low and middle-income countries (LMICs), where remote populations often lack access to specialized medical facilities [14].

InferVision's CAD for TB software, called InferRead DR Chest (InferVision, China), uses machine and deep learning algorithms to identify TB-related abnormalities within radiographic images [15]. Previous evaluations of InferRead DR Chest have reported AUCs that ranged from 0.76 - 0.85, with only one evaluation performed on African data [16–18].

The primary objective of this study was to assess the overall performance of InferRead DR Chest on data from a high HIV and TB burden setting among people being evaluated for TB. Central to this evaluation is determining an appropriate TB score that matches the WHO recommended sensitivity of at least 90% for screening tools [11].

## Methods

### Study design

A sample of anonymized observations (accessed on the 6th of June 2023) from an active case finding (ACF) study dataset conducted in Lusaka, Zambia was identified as having chest X-ray, sputum evaluation and TB diagnosis results based on Xpert or culture tests. Only observations for study participants who were 15 years and older were meant to be included in the final analysis dataset. However, the final analysis dataset also included a significant sample of observations with missing age. A chi-squared test of association analysis was conducted to ascertain whether observations with a score suggestive of TB were more or less likely to have missing data for the age variable than observations with a score not suggestive of TB. Furthermore, only observations with confirmed MTB-positive or MTB-negative results were included in the analysis; any observations with Xpert trace results were excluded.

Anonymized Chest radiography images were sent to InferVision, where they were processed using the InferRead DR Chest algorithm on their server blinded to microbiological referencing and clinical data. The authors were not involved in processing the images, and no additional training of the algorithm was required, as it was already

pre-trained. The developer's involvement was limited to providing technical support during the image processing phase, ensuring that the system functioned properly.

## Participants

The evaluation of InferRead DR Chest was based on data collected during an ACF study conducted from July 2017 to December 2018 in Lusaka, Zambia [19]. Briefly, the ACF study evaluated multi-component ACF strategies at both health facility and community levels and used both symptom screening and digital chest X-ray for TB screening, and TB diagnosis using Xpert MTB/RIF or Culture when available. Similar approaches were undertaken at both the facility and community and included demand creation through awareness raising, TB screening and diagnosis and linkage to care. Patients were screened with either symptoms or digital chest x-ray, depending on availability, followed by a definitive TB test for those who were symptomatic or who had abnormal CXR.

In the parent study, over 18,000 individuals were screened for TB through both community and facility-based approaches. 2212 radiological images from this cohort were included in our current study to evaluate the performance of InferRead DR Chest.

## Test methods

The index test used was the InferRead DR Chest CAD for TB system, which processed anonymized chest x-ray images. Each image was assigned a TB score between 0 and 1, with a higher score (closer to 1) indicating a greater likelihood of TB abnormalities. The system applied an automatic algorithm to calculate these scores.

The composite reference standard used was the Xpert MTB/RIF or MTB culture test results. The comparison of the composite reference standard test results and the index test scores facilitated the identification of true positives and true negatives.

The test positivity cutoff for the InferVision CAD system was set at TB scores that optimize the balance between sensitivity and specificity for TB detection to match the WHO recommendation of at least 90% sensitivity for screening tools.

## Data analysis

The initial analysis utilized the STATA (Stata/SE 17, StataCorp LLC) "roctab [20]" command to establish an optimal TB score by comparing the CAD software's results to the composite reference standard test results, with TB scores greater than or equal to a selected cut point identified as meeting the WHO-recommended sensitivity of at least 90%. Descriptive analysis was also performed, examining background characteristics such as sex, age, TB symptoms, history of previous TB, and HIV status in relation to the composite reference standard test results. This involved summarizing frequencies and proportions for each variable and identifying potential associations.

STATA's "diagt" command was then used to calculate the sensitivity, specificity, Negative Predictive Value (NPV), and Positive Predictive Value (PPV), while ROC curves were generated with "roccomp" to assess performance across different variables and estimate Area Under the Curve (AUC) values.

Lastly, the number needed to screen (NNS) was calculated using a direct proportion formula to estimate the number of individuals in the study population needing to be screened to identify one TB-positive case, based on the selected TB scores. This step was included as part of the analysis to quantify the screening yield of the CAD system within the context of the dataset used. Additionally, this analysis was particularly relevant for understanding the practical application and efficiency of the CAD system in facility settings, if it were implemented for active TB case finding.

## Ethical considerations

The study had ethical approval from the University of Zambia Biomedical Ethics Research Committee (UNZABREC) No: 012-05-17. Study participants provided verbal informed consent. The requirement for written informed consent was waived by the ethics committee.

## Results

In the ACF study, 18,194 individuals were screened for TB. Of the 18,194 individuals, only 10,763 had CXR; of these, only 2212 (12.2%) had submitted sputum and had complete reference test results and a CXR result. Among the 2212 observations selected, 1890 processed chest radiography images met the inclusion criteria for data analysis. Excluded from further analysis were 34 observations not meeting the age inclusion criteria (i.e., ≥ 15 years) and 288 observations lacking composite reference standard test results. However, we retained 386 observations that were missing age for further analysis. A chi-squared test of association analysis (p = 0.031) indicated that the extent of missing data for the age variable was associated with the CAD for TB score outcomes. Therefore, excluding the observations with missing age from our analysis would have biased our conclusions. The median age was 38 years, with an interquartile range (IQR) of 29–47. Observations were categorized into the following age groups: 192 (10.2%) were between 15–24, 400 (21.2%) were between 25–34, 444 (23.5%) were between 35–44, 251 (13.3%) were between 45–54 and 217 (11.5%) were 55 years and above. Regarding TB symptoms, 1730 (91.5%) participants reported experiencing at least one TB symptom (i.e., cough, weight loss, night sweats or fever). Additionally, 449 (23.8%) had a history of previous TB, and 704 (37.2%) were HIV-positive. Higher TB scores were associated with a greater likelihood of TB-related abnormalities, any score of 0.12 or above was established as suggestive of TB in this study. Therefore, of the 1890 observations, 1,234 (65.3%) were found to have TB scores greater than or equal to 0.12, while 656 (34.7%) had TB scores less than 0.12. Furthermore, it was observed that the true positive number of TB cases was 261 (21.2%), while the false positive was 973 (78.8%). In contrast, the true negative number of TB cases was 628 (95.7%), while the false negative number was 28 (4.3%). Fig 1 illustrates the data flow diagram of the observations analyzed.

Out of the total sample of 1890 observations included in the study, 289 (15.3%) were identified as TB-positive based on the composite reference standard diagnostic tests. Of the 289 observations, 214 (74.0%) were male, compared to 75 (26.0%) females, p < 0.001. The median age for individuals with TB was 34, IQR (28–42). Table 1. shows the background characteristics of observations retained for data analyses.

At a TB score cut point of 0.12, which we determined to align with the WHO's minimum sensitivity requirement of 90% for TB screening tools, InferRead DR Chest showed an overall sensitivity of 90.3% (95% CI: 86.3%—93.5%) and an overall specificity of 39.2% (95% CI: 36.8%—41.7%). The positive predictive value (PPV) and negative predictive value (NPV) of the CAD software were 21.2% (95% CI: 18.9%—23.5%) and 95.7% (95% CI: 93.9%—97.1%), respectively. Table 2. shows sensitivity, specificity, NPV, and PPV estimates stratified by each key background characteristic.

The overall AUC was 0.81 (95% CI: 0.78—0.83). For sex (p = 0.76), the AUC for females was 0.81 (95% CI: 0.75—0.87), while the AUC for males was 0.80 (95% CI: 0.76—0.83). In the age group analysis (p = 0.35), individuals aged 15–24 and 25–34 exhibited the highest AUC estimates of 0.85 (95% CI: 0.76—0.94) and 0.85 (95% CI: 0.81—0.90) respectively. In contrast, those in the age group ≥55 had the lowest AUC of 0.74 (95% CI: 0.61—0.88). HIV-positive individuals demonstrated an AUC of 0.77 (95% CI: 0.72—0.82) (p = 0.080), and individuals with a history of previous TB showed an AUC of 0.71 (95% CI: 0.63—0.79) (p = 0.003). Figs 2 and 3 show ROC curves for sex, age group, history of previous TB, and HIV status.

The overall number needed to screen to detect one TB patient from the sample retained for analysis was 6.5. Using InferRead DR Chest at a TB score cut point of 0.12, the NNS reduced to 4.7. Among individuals with CAD TB scores below the cut point of 0.12, 9.7% with TB would be missed.

## Discussion

When evaluated using data from a population with high TB and HIV prevalence, InferVision's CAD software had an overall acceptable performance (AUC 0.81). However, this performance was lower than the only reported evaluation from Africa, which observed a better performance (AUC of 0.85) [16]. The minimal difference in performance may be accounted for by

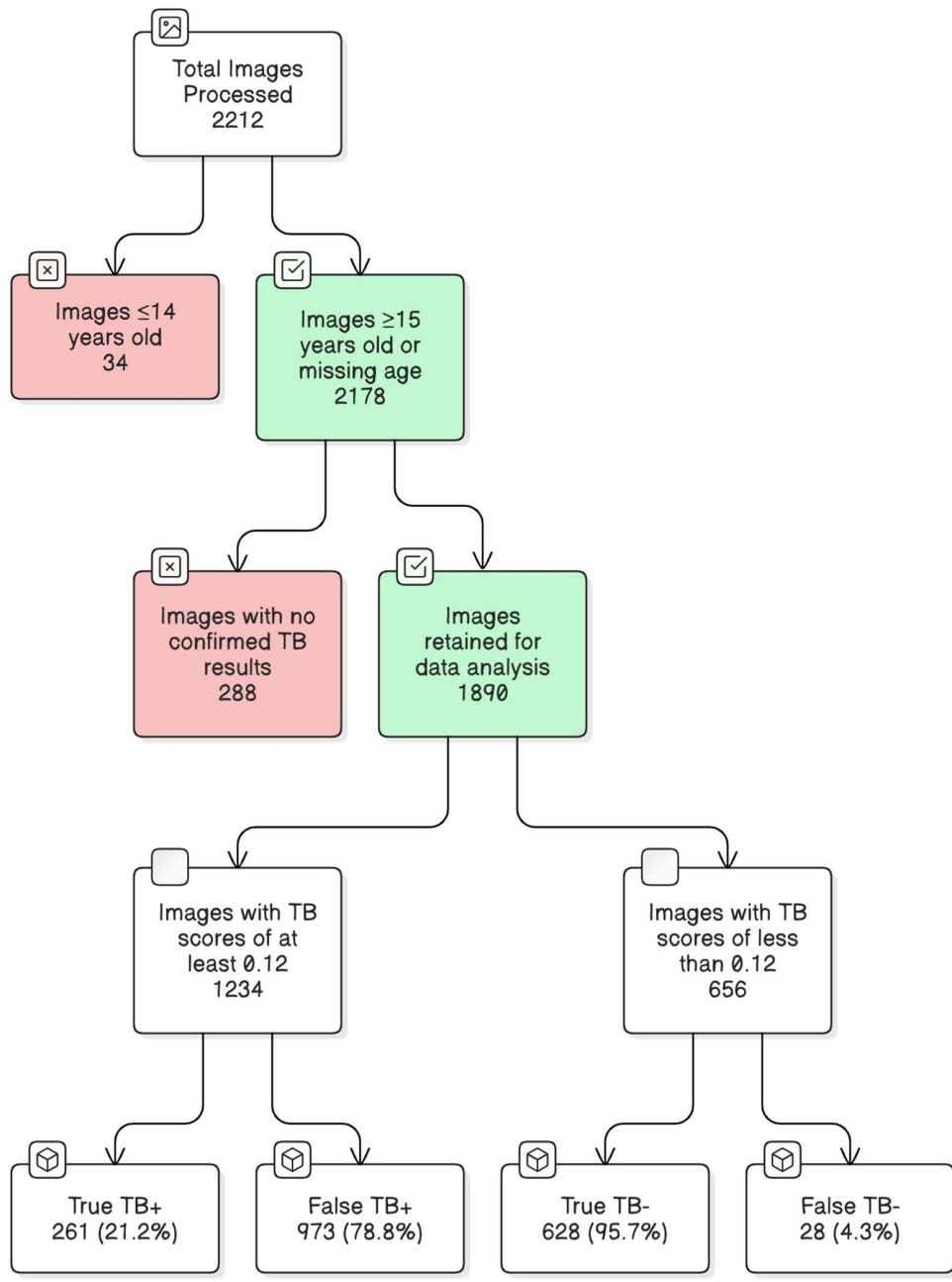

**Fig 1. Data flow diagram of the analyzed observations.**

the fact that the South African-based evaluation was on prevalence survey data, whilst our evaluation was performed on data collected in an active case finding environment. Worthy of note is the difference in HIV prevalence, 18% and 37% for the South African study and our study, respectively. Many evaluations of different CAD software, including InferRead DR Chest have observed poor performance in HIV positive individuals [21].

When compared to CAD4TB and qXR performance on the same population, InferRead DR Chest did not perform as well. Both CAD4TB and qXR demonstrated overall excellent performance, as seen from the AUCs of ≥85 [21].

**Table 1. Background characteristics of observations retained for data analyses.**

| | Total number of eligible participants N (% of total) | TB xpert or culture test result | |
| --- | --- | --- | --- |
| | | Negative n (%) | Positive n (%) |
| **Overall** | **1890 (100)** | **1601 (84.7)** | **289 (15.3)** |
| **Sex** | | | |
| Female | **704 (37.2)** | 629 (39.3) | 75 (26.0) |
| Male | **1186 (62.8)** | 972 (60.7) | 214 (74.0) |
| **Age group** | | | |
| Median (IQR) | **38 (29-47)** | 39 (30-48) | 34 (28-42) |
| 15-24 | **192 (10.2)** | 157 (9.8) | 35 (12.1) |
| 25-34 | **400 (21.2)** | 309 (19.3) | 91 (31.5) |
| 35-44 | **444 (23.5)** | 380 (23.7) | 64 (22.1) |
| 45-54 | **251 (13.3)** | 219 (13.7) | 32 (11.1) |
| 55+ | **217 (11.5)** | 198 (12.4) | 19 (6.6) |
| Missing | **386 (20.4)** | 338 (21.1) | 48 (16.6) |
| **Atleast one TB symptom** | | | |
| No | **159 (8.4)** | 146 (9.1) | 13 (4.5) |
| Yes | **1730 (91.5)** | 1454 (90.8) | 276 (95.5) |
| Missing | **1 (0.1)** | 1 (0.1) | 0 (0.0) |
| **History of previous TB** | | | |
| No | **1436 (76.0)** | 1206 (75.3) | 230 (79.6) |
| Yes | **449 (23.8)** | 391 (24.4) | 58 (20.1) |
| Missing | **5 (0.3)** | 4 (0.2) | 1 (0.3) |
| **HIV status** | | | |
| Negative | **1076 (56.9)** | 923 (57.7) | 153 (52.9) |
| Positive | **704 (37.2)** | 587 (36.7) | 117 (40.5) |
| Indeterminate | **1 (0.1)** | 1 (0.1) | 0 (0.0) |
| Missing | **109 (5.8)** | 90 (5.6) | 19 (6.6) |

Using the selected TB score cut point of 0.12, InferRead DR Chest had significantly lower specificity in males than females. Additionally, the low specificity of 39.2% highlighted the software's lack of precision in correctly excluding individuals without TB and is far below the recommended specificity of 70% for screening tools. We recognize that adjusting the TB scores would lead to changes in both sensitivity and specificity, and the selected TB score was chosen to prioritize sensitivity in alignment with WHO CAD for TB screening tool requirements. Additionally, the software attained a high sensitivity at a low TB score, suggesting a minimal potential to reduce the NNS and meet anticipated efficiencies if used as a screening tool. In settings where the CAD tool is used for mass TB screening, a higher rate of false positives would necessitate additional confirmatory diagnostic tests. This could increase healthcare costs and resource utilization, particularly in low-resource settings where such testing is expensive.

Despite the increased costs, the use of CAD for initial screening may still be justified, particularly in high-burden areas, as it ensures that fewer true TB cases are missed. However, careful consideration must be given to balancing sensitivity and specificity to optimize both cost-effectiveness and the accuracy of TB detection. Future research should focus on refining TB score thresholds for different populations or settings to mitigate the economic and operational burden of false positives while maintaining robust screening performance.

**Table 2. Sensitivity, specificity, PPV, and NPV estimates are stratified by each key background characteristic.**

| | Sensitivity % (95% CI) | Specificity % (95% CI) | PPV % (95% CI) | NPV % (95% CI) |
|---|---|---|---|---|
| **Overall** | **90.3% (86.3%-93.5%)** | **39.2% (36.8%-41.7%)** | **21.2% (18.9%-23.5%)** | **95.7% (93.9%-97.1%)** |
| **Sex** | | | | |
| Female | 85.3 (75.3-92.4) | 47.9 (43.9-51.8) | 16.3 (12.8-20.4) | 96.5 (93.8-98.2) |
| Male | 92.1 (87.6-95.3) | 33.6 (30.7-36.7) | 23.4 (20.6-26.4) | 95.1 (92.2-97.1) |
| **Age group*** | | | | |
| 15-24 | 85.7 (69.7-95.2) | 60.5 (52.4-68.2) | 32.6 (23.2-43.2) | 95.0 (88.7-98.4) |
| 25-34 | 92.3 (84.8-96.9) | 45.0 (39.3-50.7) | 33.1 (27.3-39.2) | 95.2 (90.4-98.1) |
| 35-44 | 90.6 (80.7-96.5) | 37.1 (32.2-42.2) | 19.5 (15.2-24.5) | 95.9 (91.3-98.5) |
| 45-54 | 90.6 (75.0-98.0) | 33.3 (27.1-40.0) | 16.6 (11.4-22.9) | 96.1 (88.9-99.2) |
| 55+ | 84.2 (60.4-96.6) | 34.3 (27.8-41.4) | 11.0 (6.4-17.2) | 95.8 (88.1-99.1) |
| **Atleast one TB symptom*** | | | | |
| No | 61.5 (31.6-86.1) | 44.5 (36.3-53.0) | 9.0 (4.0-16.9) | 92.9 (84.1-97.6) |
| Yes | 91.7 (87.8-94.6) | 38.7 (36.2-41.3) | 22.1 (19.7-24.6) | 96.1 (94.2-97.5) |
| **Previous history of TB*** | | | | |
| No | 90.0 (85.4-93.6) | 45.9 (43.0-48.7) | 24.1 (21.2-27.1) | 96.0 (94.1-97.5) |
| Yes | 91.4 (81.0-97.1) | 18.9 (15.2-23.2) | 14.3 (10.9-18.3) | 93.7 (85.8-97.9) |
| **HIV status*** | | | | |
| Negative | 91.5 (85.9-95.4) | 40.2 (37.0-43.4) | 20.2 (17.3-23.4) | 96.6 (94.3-98.2) |
| Positive | 88.9 (81.7-93.9) | 36.6 (32.7-40.7) | 21.8 (18.2-25.8) | 94.3 (90.4-96.9) |

*Variable missing data excluded from sensitivity, specificity, NPV, and PPV analysis.

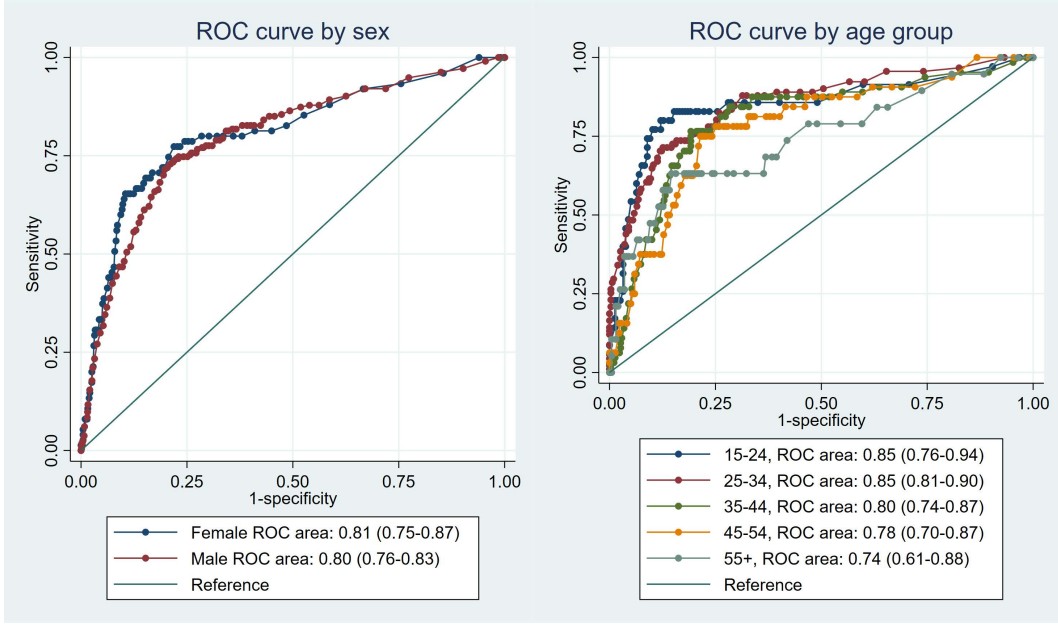

**Fig 2. ROC curves for sex and age group.**

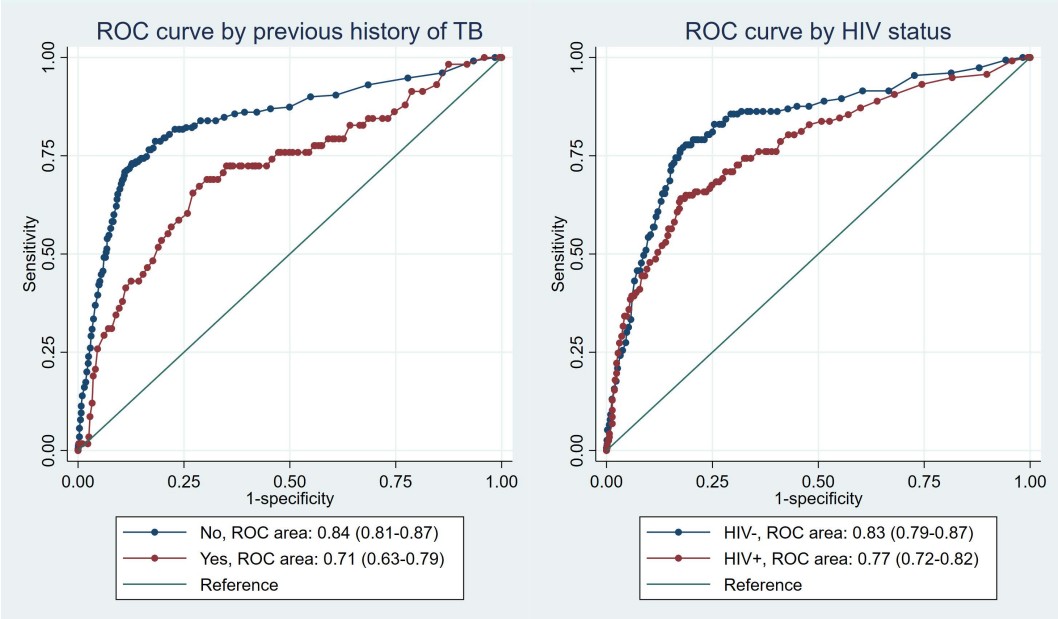

**Fig 3. ROC curves for history of previous TB and HIV status.**

Furthermore, no significant difference in AUC estimates was observed between males and females. Similar findings for both sexes, with overlapping confidence intervals, suggest that the CAD software's performance in detecting TB was consistent regardless of sex. This finding underlines the software's impartiality in its diagnostic accuracy, making it equally effective for both male and female populations. In contrast, notable differences were observed in the CAD software's performance between individuals aged 45 years and above and those below the age of 45. The AUC estimates revealed distinct patterns, suggesting that the diagnostic accuracy of the CAD software may vary across different age groups, emphasizing the importance of considering age-related factors when implementing and interpreting the results. Similarly, there was a significant difference in the AUC estimates between individuals who had a history of previous TB (or an HIV-positive status). The CAD software exhibited a notably "better" performance in individuals without a history of previous TB (or individuals with an HIV-negative status), like what has been observed by other studies [17,21]. These findings emphasize the importance of considering the patient's medical history when using the software for TB screening and suggests the need to train this software with data enhanced for these subpopulations [22].

The strength of our study is that we used data from an active case finding setting. The findings likely reflect the program setting. Additionally, this is the first study to evaluate InferRead DR Chest in a population with high HIV prevalence.

However, the study has some limitations. Most of the participants in this study were symptomatic, meaning we cannot draw conclusions about the CAD tool's ability to detect sub-clinical TB. Furthermore, we acknowledge the limitations of Xpert MTB/RIF, including its potential for false positives and false negatives, despite being widely used for TB diagnosis. Therefore, diagnostic challenges may influence the evaluation of InferRead DR Chest and should be considered when interpreting the system's performance. Additionally, some observations were missing age, preventing the conduct of sensitivity, specificity, PPV, and NPV analyses for these observations.

## Conclusion

InferRead had an overall acceptable performance. Poor performance was observed among people with prior TB or have an HIV positive status. There were significant differences in performance among those aged older than 45 years, showing a trend towards reduced performance as age increases. Our findings suggest that additional training of InferRead Chest DR must be considered before it can be used for TB screening in our population.

## Supporting information

**S1 File. Additional files detailing specific data analysis procedures and results outputs, including thresholds, diagnostic performance metrics, and subgroup evaluations to support the findings presented in the manuscript.** (ZIP)

## Acknowledgments

We extend our heartfelt gratitude to all those involved in the ACF study, particularly the participants whose invaluable involvement made this research possible. We extend our deepest appreciation to those whose chest X-rays were used to evaluate InferRead DR Chest; their contribution marks a pivotal step in advancing tuberculosis diagnosis. The authors would also like to thank InferVision, which processed the sampled chest X-ray images at no cost.

## Author contributions

**Conceptualization:** Monde Muyoyeta.

**Data curation:** Paul Somwe, Minyoi Maimbolwa, Kanema Chiyenu.

**Formal analysis:** Paul Somwe.

**Funding acquisition:** Monde Muyoyeta.

**Investigation:** Monde Muyoyeta.

**Methodology:** Monde Muyoyeta.

**Resources:** Monde Muyoyeta.

**Software:** Kanema Chiyenu.

**Supervision:** Monde Muyoyeta.

**Validation:** Monde Muyoyeta.

**Writing – original draft:** Paul Somwe.

**Writing – review & editing:** Paul Somwe, Mwansa Lumpa, Mary Kagujje, Monde Muyoyeta.

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
