## [Decision Letter · Decision Letter 0]

PGPH-D-24-01483

Evaluating the Performance of InferVision’s Computer-Aided Detection (CAD) Algorithm for Tuberculosis (TB) Screening

Dear Dr. Somwe,

Thank you for submitting your manuscript to PLOS Global Public Health. After careful consideration, we feel that it has merit but does not meet PLOS Global Public Health’s publication criteria as it currently stands. Therefore, we invite you to submit a revised version of the manuscript that addresses the points raised during the review process.

I agree with the reviewers that there is value in the data reported in this manuscript but that the analysis is unclear and the way it is reported is confusing. To be accepted this manuscript will need a major revision that addresses all of the reviewer comments thoroughly. It is especially important to address the concern of Reviewer #1 that questions how the threshold was selected. Please address this with a complete explanation in the methods and report of results in the results section. 

Please note that Reviewer #2's specific comments are included as an attachment. In your rebuttal letter please import these into a single rebuttal document.

We look forward to receiving your revised manuscript.

Kind regards,

Emily B. Wong

Academic Editor

Journal Requirements:

Additional Editor Comments (if provided):

Reviewers' comments:

Reviewer's Responses to Questions

**Comments to the Author**

1. Does this manuscript meet PLOS Global Public Health’s publication criteria ? Is the manuscript technically sound, and do the data support the conclusions? The manuscript must describe methodologically and ethically rigorous research with conclusions that are appropriately drawn based on the data presented.

Reviewer #1: Yes

Reviewer #2: Yes

2. Has the statistical analysis been performed appropriately and rigorously?

Reviewer #1: Yes

Reviewer #2: Yes

3. Have the authors made all data underlying the findings in their manuscript fully available (please refer to the Data Availability Statement at the start of the manuscript PDF file)?

Reviewer #1: Yes

Reviewer #2: No

4. Is the manuscript presented in an intelligible fashion and written in standard English?

Reviewer #1: Yes

Reviewer #2: Yes

5. Review Comments to the Author

Reviewer #1: I read with interest this study that aimed to evaluate the performance of a CAD software (InferRead DR Chest) and to determine an optimal threshold for TB detection. The authors report that at a threshold of 10, sensitivity was 94.1% while specificity was 26.9%. The authors further note differences in accuracy between sub-groups. They conclude by stating that additional training/piloting needs to be considered for this CAD software before it can be used for TB screening.

Overall impression:

This study adds to the limited data evaluating InferRead DR Chest software. However, there are methodological limitations and a confusing manner in which the authors present their findings. For example, the authors aim to report the optimal threshold, yet there is uncertainty on how or why that threshold was chosen. Furthermore, there is no clarity on the specific population included nor the reference standard used to assess CAD accuracy. This is essentially a diagnostic accuracy study, and the authors may benefit from following the STARD guidelines when presenting their study. Furthermore, although one can decipher what the authors meant in their conclusion, it could be more consistent with the primary objective of the study. Finally, the authors should review the manuscript to address grammatical and typographical errors. Below are specific comments and suggestions.

Introduction:

The opening paragraphs describe the global challenges of TB detection and the need for additional tools. However, the authors should be careful not to infer that CAD is a TB diagnostic. CAD has been recommended by the WHO for screening/triage testing. CAD cannot diagnose TB alone and therefore microbiological diagnostic testing (e.g., NAAT, culture) is still required. It is suggested the authors ensure this important point is reflected.

Line 76. Although noted that InferRead DR Chest is from InferVision, it is recommended that it be written ‘InferRead DR Chest (InferVision, China)’.

The authors note that the primary objective of this manuscript was to assess the overall performance of InferRead DR Chest, including a threshold that optimizes sensitivity. However, the authors do not highlight any previous literature that has evaluated this CAD software. This is important and sets the stage for the reader, assisting them to know what has already been reported and how this study differs and adds to the evidence base.

Methods:

As a diagnostic accuracy study. The authors are encouraged to present their methodology according to STARD guidelines: (i) Bossuyt PM, Reitsma JB, Bruns DE, Gatsonis CA, Glasziou PP, Irwig L, et al, For the STARD Group. STARD 2015: An Updated List of Essential Items for Reporting Diagnostic Accuracy Studies, (ii) https://bmjopen.bmj.com/content/6/11/e012799).

Lines 86-87. The authors report that radiological images from the TB REACH Wave 5 study were evaluated by InferRead DR Chest. It would be helpful to briefly summarise this study in the current manuscript in addition to citing the study. Furthermore, what did Wave 5 entail? How many were screened in the parent study and how many of those were included in the current study?

Lines 87-89. The authors report ‘…multi-component active case-finding study conducted similar activities at the health facility and the community’. The population included is a critical aspect of the paper and is currently confusing for the reader. The authors should clarify exactly what population was studied as CAD performance can vary between active and passive case-finding. If patients were recruited from both community and facility settings, the authors should describe this and should split the analysis between these distinct populations.

Line 103. Further details are required about the CAD analysis. For example, were images sent to InferVision for analysis? Was it done by the authors on a server provided by InferVision? Was there any training involved? To what extent were the developers involved?

The data analysis section could be refined and truncated.

Line 110. The full name, company, and version of the statistical software used should be reported.

Lines 115-117. As one of the main aims of the study, the identification of the optimal threshold should not be reported in the methods, but rather the results section along with the threshold’s corresponding sensitivity and specificity.

I am not convinced that the analysis for the NNS is relevant or sound in this analysis. Please can the authors clarify their rationale and provide specific details in the data analysis section as it is currently limited on details.

Results

It would be more readable if the authors presented their results in a more systematic manner. For example, report the included participants and exclusions, then demographics, then threshold(s) and accuracy estimates.

Lines 161-162. Please could the authors provide more details on the sensitivity analysis. Perhaps add as a supplement?

To improve readability and understanding, it is suggested that the authors report the total number of patients screened in the parent study and the percentage of those included in the current manuscript. Furthermore, it would be helpful to report the number of TB-positive individuals in the study in the text, as opposed to only in Table 1.

Fig 1. The TP, FP, TN, and FN would fit better if reported with the accuracy estimates lower down in the results section.

Table 1. The authors report ‘TB Xpert or culture test result’ but only Xpert in the text of the methods section. Can the authors please clarify what the reference standard was. This is critical as the interpretation of CAD accuracy will be different.

Table 1. The authors have provided percentages confusingly. It would be easier for the reader if the percentage were out of the TB-negative and TB-positive groups per column (i.e., out of 1601 and out of 289, respectively).

Lines 174-178 and Table 2. Although I can vaguely understand what the authors are trying to report, the way in which the authors present their performance results is rather confusing. One cannot say ‘an overall sensitivity of 94.1%’ as sensitivity will change depending on your threshold. It appears the authors have set the threshold first and then reported the corresponding sensitivity and specificity. In a retrospective analysis such as this, it would make more sense to show the different sensitivities and specificities across multiple thresholds and then determine the optimal threshold (e.g., a threshold that corresponds to 90% sensitivity). Furthermore, why was 10 chosen as the optimal threshold with a sensitivity of 94.1%? Was there a slightly higher threshold that had ~90% sensitivity but improved specificity?

The threshold cannot be ‘>10’ – this is confusing. A CAD threshold is a single number resulting in a binary classification (i.e., positive if above and negative if below).

Lines 181-189 and Fig 2. It is clinically relevant and strengthens the study to have sub-group analyses. The authors show AUCs were lower for males, older individuals, those with a history of previous TB, and people living with HIV. These findings corroborate findings of previous studies. However, did the authors compare AUCs statistically? Were there significant differences?

Discussion:

Line 204-205. The authors may want to do an updated literature search as there are several publications that have evaluated InferRead for TB screening. For example, (i) Codlin AJ, et al. Independent evaluation of 12 artificial intelligence solutions for the detection of tuberculosis. Sci Rep. 2021;11(1):23895; (ii) Qin ZZ, et al. Tuberculosis detection from chest x-rays for triaging in a high tuberculosis-burden setting: an evaluation of five artificial intelligence algorithms. Lancet Digit Health. 2021;3(9):e543-e554; (iii) Qin ZZ, et al. Computer-aided detection of tuberculosis from chest radiographs in a tuberculosis prevalence survey in South Africa: external validation and modelled impacts of commercially available artificial intelligence software. Lancet Digit Health. 2024:S2589-7500(24)00118-3.

Lines 212-217. The authors may want to reword their point on CAD ‘demonstrating good sensitivity’. Please remember that you have retrospectively set the threshold to report that sensitivity (with a corresponding low specificity of ~26.9%). If you changed the threshold, your sensitivity and specificity would also change.

Lines 222-241. The authors report statistical significance between AUCs in the discussion but make no mention of this in the results (no p-values, etc.) nor in the data analysis section in the methods. See comment above and please clarify.

It would be beneficial for the authors to expand on the clinical implications of the findings. For example, having a low specificity (therefore, many false positives) will result in increased costs for additional diagnostic microbiological testing.

It is suggested that the authors compare their findings to similar studies and discuss similarities and differences in terms of methods, results, etc.

Lines 242-247. The authors should re-evaluate their strengths and limitations section. For strengths, the authors report a ‘substantial sample size’ yet have not included sufficient detail about the population studied (e.g., active vs passive case-finding). The authors may want to include having a microbiological reference standard as a strength. For limitations, the authors may need to consider noting that Xpert MTB/RIF was used and not the more sensitive Ultra which may influence accuracy estimates of CAD. Furthermore, the authors may want to add that they may not be able to comment on CAD accuracy to detect sub-clinical TB (as >90% of participants were symptomatic).

Conclusion

Although one can work out what the authors conclude, it is suggested that the authors focus on aligning the primary aim and their findings in the conclusion (e.g., InferRead DR Chest performance did not meet expectations / it did not meet WHO TPP / at the optimal threshold (sensitivity 90%) specificity was low, etc.).

The authors bring in new ideas into the conclusion which may be better suited to the discussion section.

Abstract

The abstract could be more concisely written, especially the methods section. This could be truncated to allow more findings to be reported in the results section. The authors may want to amend the abstract after reviewing the comments above.

Additional comments

The funding source, ethical considerations, and potential conflicts of interest were reported adequately.

Reviewer #2: The manuscript is well written in standard English and presented in an intelligible manner. Statistical analysis is well conducted, and the results are well presented. The discussion and conclusion are well convincing. The dataset is not available though, for public use except upon requesting it from the lead author. Minor revision is required to improve the quality of the manuscript.

6. PLOS authors have the option to publish the peer review history of their article (what does this mean? ). If published, this will include your full peer review and any attached files.

**Do you want your identity to be public for this peer review?** For information about this choice, including consent withdrawal, please see our Privacy Policy .

Reviewer #1: No

Reviewer #2: **Yes: ** Alfred Keter

---

## [Decision Letter · Decision Letter 1]

PGPH-D-24-01483R1

Evaluating InferVision’s Computer-Aided Detection (CAD) Algorithm for Tuberculosis (TB) Screening, Lusaka Zambia

Dear Dr. Somwe,

Thank you for submitting your manuscript to PLOS Global Public Health. After careful consideration, we feel that it has merit but does not fully meet PLOS Global Public Health’s publication criteria as it currently stands. Therefore, we invite you to submit a revised version of the manuscript that addresses the points raised during the review process.

Please note the important remaining concerns that Reviewer 1 has listed. It will be necessary to address each of these with a comprehensive and complete response in order for the manuscript to be accepted. In particular please structure and provide clear explanation for the information contained within the supplementary materials.

Please note that Reviewer 2 has also suggested several important points to address in the attached letter.

Please copy and paste the concerns of both reviewers and your responses into a single rebuttal letter to ensure that the next round of review is efficient.

We look forward to receiving your revised manuscript.

Kind regards,

Emily B. Wong

Academic Editor

Journal Requirements:

Additional Editor Comments (if provided):

Reviewers' comments:

Reviewer's Responses to Questions

**Comments to the Author**

1. If the authors have adequately addressed your comments raised in a previous round of review and you feel that this manuscript is now acceptable for publication, you may indicate that here to bypass the “Comments to the Author” section, enter your conflict of interest statement in the “Confidential to Editor” section, and submit your "Accept" recommendation.

Reviewer #1: (No Response)

Reviewer #2: All comments have been addressed

2. Does this manuscript meet PLOS Global Public Health’s publication criteria ? Is the manuscript technically sound, and do the data support the conclusions? The manuscript must describe methodologically and ethically rigorous research with conclusions that are appropriately drawn based on the data presented.

Reviewer #1: Yes

Reviewer #2: Yes

3. Has the statistical analysis been performed appropriately and rigorously?

Reviewer #1: Yes

Reviewer #2: Yes

4. Have the authors made all data underlying the findings in their manuscript fully available (please refer to the Data Availability Statement at the start of the manuscript PDF file)?

Reviewer #1: Yes

Reviewer #2: Yes

5. Is the manuscript presented in an intelligible fashion and written in standard English?

Reviewer #1: Yes

Reviewer #2: Yes

6. Review Comments to the Author

Reviewer #1: I thank the authors for resubmitting this manuscript evaluating the diagnostic performance of InferRead DR Chest for TB screening. The authors have addressed most comments. However, I have several comments remaining.

Comment: I am still confused as to why the threshold is being stated in the methods section. Part of the aim of this study was to evaluate for an optimal threshold that matched a sensitivity of 90%. The authors are therefore stating results in the methods section. Can the authors clearly explain why they chose to report this finding/result in the methods section.

Comment: Although the authors respond to the original comment, can the authors please clearly explain why they still use the ‘greater than or equal (≥)’ symbol for the threshold, as it is still confusing to me. A threshold is a single number between 0 and 1 (or 0 and 100) corresponding to a certain sensitivity and specificity. It cannot be above or below that number as the respective sensitivity and specificity would change. Currently the authors state that “At 94.1% sensitivity, InferRead DR Chest had a specificity of 26.9% at a TB score threshold of ≥0.1”. It cannot be greater than or equal to 0.1, it is 0.1. Anything above (or below) 0.1 would have a different sensitivity and specificity.

Comment: Please can the authors note how they managed Xpert trace results. I did not find mention of it in the manuscript.

Comment: I appreciate the authors adding a brief description of the parent study. However, please can the authors mention why only ~2,000 X-rays of the 18,000 individuals screened were included in the current study. Were these selected? If so, how? Were these the only images available?

Comment: My previous comment on whether there were slightly higher thresholds that had ~90% sensitivity but improved specificity was not completely addressed. Although I received the supplementary material, these were image files and only had thresholds ranging from 0.0479 to 0.102305. Where are the other thresholds? For example, what were the sensitivities and specificities of thresholds between 0.1 and 0.2?

Comment: I would like to see a more structured supplementary material. I do not believe several image files is sufficient.

Reviewer #2: This has been uploaded as a separate document.

7. PLOS authors have the option to publish the peer review history of their article (what does this mean? ). If published, this will include your full peer review and any attached files.

**Do you want your identity to be public for this peer review?** For information about this choice, including consent withdrawal, please see our Privacy Policy .

Reviewer #1: No

Reviewer #2: **Yes: ** Alfred Keter

---

## [Editor Report · Decision Letter 2]

Evaluating InferVision’s Computer-Aided Detection (CAD) Algorithm for Tuberculosis (TB) Screening, Lusaka Zambia

PGPH-D-24-01483R2

Dear Mr Somwe,

We are pleased to inform you that your manuscript 'Evaluating InferVision’s Computer-Aided Detection (CAD) Algorithm for Tuberculosis (TB) Screening, Lusaka Zambia' has been provisionally accepted for publication in PLOS Global Public Health.

**** Important note**** I can tell that the reviewers comments were adequately responded to by referring to the "track changes" manuscript in the most recent submission. However, the "clean" version in my submission appears to be the original version of the manuscript (prior to any peer review). Please ensure that the correct "clean" version of the manuscript is submitted. *****

Best regards,

Emily B. Wong

Academic Editor